# Early Biomarkers of Neurodegenerative and Neurovascular Disorders in Diabetes

**DOI:** 10.3390/jcm9092807

**Published:** 2020-08-30

**Authors:** Aleksandra Gasecka, Dominika Siwik, Magdalena Gajewska, Miłosz J. Jaguszewski, Tomasz Mazurek, Krzysztof J. Filipiak, Marek Postuła, Ceren Eyileten

**Affiliations:** 11st Chair and Department of Cardiology, Medical University of Warsaw, 02-097 Warsaw, Poland; dominika.siwik@gmail.com (D.S.); gmgajewska@gmail.com (M.G.); tmazurek@kardia.edu.pl (T.M.); krzysztof.filipiak@wum.edu.pl (K.J.F.); 21st Department of Cardiology, Medical University of Gdansk, 80-211 Gdansk, Poland; jamilosz@gmail.com; 3Department of Experimental and Clinical Pharmacology, Centre for Preclinical Research and Technology, Medical University of Warsaw, 80-211 Warsaw, Poland; mpostula@wum.edu.pl (M.P.); ceyileten@wum.edu.pl (C.E.)

**Keywords:** diabetes mellitus complications, neurovascular, neurodegenerative, stroke, dementia, biomarkers, microRNA, growth factors, oxidative stress, PD, AD, MCI, DM

## Abstract

Diabetes mellitus (DM) is a common disease worldwide. There is a strong association between DM and neurovascular and neurodegenerative disorders. The first group mainly consists of diabetic retinopathy, diabetic neuropathy and stroke, whereas, the second group includes Alzheimer’s disease, Parkinson’s disease, mild cognitive impairment and dementia. The aforementioned diseases have a common pathophysiological background including insulin resistance, oxidative stress, atherosclerosis and vascular injury. The increasing prevalence of neurovascular and neurodegenerative disorders among diabetic patients has resulted in an urgent need to develop biomarkers for their prediction and/or early detection. The aim of this review is to present the potential application of the most promising biomarkers of diabetes-related neurodegenerative and neurovascular disorders, including amylin, β-amyloid, C-reactive protein (CRP), dopamine, gamma-glutamyl transferase (GGT), glycogen synthase kinase 3β, homocysteine, microRNAs (mi-RNAs), paraoxonase 1, phosphoinositide 3-kinases, tau protein and various growth factors. The most clinically promising biomarkers of neurovascular and neurodegenerative complications in DM are hsCRP, GGT, homocysteine and miRNAs. However, all biomarkers discussed in this review could become a part of the potential multi-biomarker screening panel for diabetic patients at risk of neurovascular and neurodegenerative complications.

## 1. Introduction

Diabetes mellitus (DM) is a common disease worldwide. Currently, it is estimated that around 425 million patients suffer from DM, and taking into account the rapidly increasing prevalence of DM in the past 40 years, that number may hit 640 million by 2040 [1]. DM has gained a lot of attention due to its association with multiple organ complications, which lead to premature function impairment and increased mortality [2]. Among them, neurodegenerative and neurovascular complications have recently become a focus of intensive research. These disorders are often suggested to have the same origin, including insulin resistance, oxidative stress, atherosclerosis, vascular injury and more. Hence, an urgent need for biomarkers of diabetic complications has arisen. The most promising early biomarkers to predict and/or early detect neurodegenerative and neurovascular complications in diabetes include C-reactive protein (CRP), microRNAs (miRNAs), paraoxonase 1 (PON-1), tau protein, β-amyloid, glycogen synthase kinase 3β (GSK-3β), phosphoinositide 3-kinases (PI3K), amylin, dopamine, gamma-glutamyl transferase (GGT), various growth factors and homocysteine.

The aim of this review is to present the potential application of these biomarkers in early detection of diabetes-related neurodegenerative and neurovascular disorders. The disorders which are the focus of this review are summarized in Table 1. Among neurodegenerative disorders, we discuss Alzheimer’s disease (AD), Parkinson’s disease (PD), mild cognitive impairment (MCI) and dementia. Among neurovascular disorders, we focus on diabetic retinopathy (DR), diabetic neuropathy and stroke. The pathophysiological mechanisms linking diabetes with neurodegenerative and neurovascular disorders are presented in Figure 1. Because of the cross-talk between the mechanisms underlying their development, we present the potential biomarkers for their early detection per biomarker, not per disease.

## 2. C-Reactive Protein

CRP is a commonly known acute phase protein produced by the liver in response to inflammation and interleukin (IL)-6. CRP is elevated in many diseases which have an inflammatory background, including DM. The recently introduced high sensitivity CRP (hsCRP) assay is more accurate than a classical CRP assay to detect subclinical inflammation [3]. Increased concentration of hsCRP is associated with higher risk of type 2 diabetes mellitus (T2DM) [4]. Since the concentration of hsCRP positively correlates with insulin resistance, hsCRP might potentially be used to predict the development of T2DM in patients with insulin resistance [5]. Increased concentration of hsCRP has also been found in many neurodegenerative and neurovascular disorders, including AD, PD, MCI and age-related macular degeneration (AMD) [6,7,8].

The link between the inflammatory state, vascular disease and cognitive malfunction is based on a hypothesis that permanently increased glucose concentration leads to microvascular changes, which cause chronic hypoperfusion of the brain and subsequent degeneration and death of the brain cells. The microvascular changes trigger low-grade inflammatory response and further aggravate the risk of neurodegeneration and neurovascular complications (Figure 1) [9]. Concurrently, various hypoglycemic agents have been found to exert anti-inflammatory effects and decrease serum concentration of hsCRP, including metformin, peroxisome proliferator-activated receptor-γ agonists, dipeptidyl peptidase-4 inhibitors, glucagon-like peptide-1 receptor agonists and insulin [10]. On the other hand, there is evidence that metformin along with other antidiabetic drugs may prevent the development of DM-associated neurodegenerative complications by balancing the survival and death signaling in cells, thus avoiding the neuronal death that defines neurodegenerative disease [11].

Some studies suggest that elevated CRP concentration in patients with DM is correlated with the risk of AD development [12]. However, according to a meta-analysis of 10 cross-sectional studies including 2093 patients, no difference in the serum CRP concentrations was observed between controls and patients with AD [12]. Neuroinflammation is also seen as a key factor in PD pathogenesis. Accordingly, increased hsCRP concentrations in serum and cerebrospinal fluid (CSF) were observed in PD patients, compared to healthy controls [13]. Further, in course of AMD, choroidal vascular dysfunction of the endothelial cell results in tissue injury, which is associated with hsCRP increase [12]. However, further studies are needed to show a direct link between DM and the above-mentioned neurodegenerative and neurovascular disorders. A fundamental part of these studies should differentiate whether inflammation and increased hsCRP is a cause, or a result of the disease, and whether it is associated with neurodegenerative and neurovascular complications in DM [12] (Table 2).

### 2.1. MicroRNAs

MiRNAs are short (19 to 25 nucleotides) non-coding RNA molecules which play a key role in post-transcriptional gene expression and regulate multiple physiological functions [81]. Several different methods were used in order to detect miRNAs, including Northern blotting, microarray analysis, quantitative real-time polymerase chain reaction (qRT-PCR) and next-generation sequencing (NGS). Apart from these techniques, new strategies have been developed, including sensitive miRNA biosensors such as silicon nanowires, gold nanoparticles, silver nanoclusters and conducting polymer/carbon nanotube hybrids [82,83,84,85,86].

MiRNAs are known to participate in maintaining not only glucose, but also subcutaneous adipose tissue homeostasis, and they might be involved in the pathogenesis of T2DM [87]. For example, expression of five different miRNAs-miR (miR-661, miR-571, miR-770-5p, miR-892b and miR-1303) was higher in 184 T2DM patients, compared to 92 healthy controls [88]. In addition, the expression levels of the above-mentioned miRNAs were upregulated in T2DM patients with diabetic nephropathy, neuropathy and retinopathy, compared to those without diabetic complications [82].

DR is one of the most extensively studied diabetic complications, since it is a major cause of blindness and vision impairment among adults. Plasma levels of miR-320a are lower in patients with DR compared to patients with DM and no retinopathy and to healthy controls [89]. MiR-320 is also involved in the regulation of insulin growth factor-1 (IGF-1) expression, which takes part in the development of insulin resistance in adipose tissue and endothelial cells [89]. It is noteworthy that patients with DR presented with higher risk of AD compared to healthy controls, indicating the association between dysregulated miRNAs, DM, DR and AD [90].

Currently, there are many studies searching for miRNA-based biomarkers of AD. In a systematic review of 20 studies it was reported that as much as 102 circulating miRNAs were dysregulated in patients with AD compared to healthy controls [91]. MiRNAs regulate the expression of various genes involved in AD development, such as presenilin, beta-secretase 1, amyloid precursor protein (APP), translocase of outer mitochondrial membrane 40 (TOM40) and brain-derived neurotrophic factor (BDNF) [92]. The most promising miRNA associated with AD include miR-455-3p and miR-34a-5p, miR-93, miR-501-3p, miR-98, miR-124, miR-92a-3p, miR-181c-5p and miR-210-3p [92,93]. When it comes to PD, miR-133b and miR-184 were found to be downregulated in this group of patients [92]. In patients with diabetic neuropathy, in turn, dysregulation of miR-379-5p was observed. In addition, miR-203, miR-96 and miR-7a expressions in dorsal root ganglion were connected with neuropathic pain, whereas miR-128a and miR-146a were connected with diabetic polyneuropathy [94].

There is evidence that metformin affects the expression of at least 13 different miRNAs, thus supporting the development of miRNA-based therapeutic strategies against DM [85]. On the other hand, metformin has been shown to upregulate miR-34a, which plays an important role in responding to DNA damage and protecting against neurodegeneration, potentially providing the combined anti-diabetic and anti-neurodegenerative benefits [86].

The above-mentioned dysregulations of miRNAs found in patients suffering from AD, DR, diabetic neuropathy and diabetic nephropathy and their modulation by anti-diabetic agents are promising fields of biomarker research. However, more detailed studies are needed to explore the potential of individual miRNAs in neurovascular and neurodegenerative complications of diabetes.

### 2.2. Paraoxonase 1

PON1 is a glycoprotein with antioxidant and anti-inflammatory effects. Due to its affinity to high-density lipoproteins, PON1 also has antiatherogenic and antioxidative properties [15]. The gene of PON1 shows substantial polymorphism, with at least eight different single-nucleotide polymorphisms identified [15]. The genetic polymorphism of PON1 and its serum activity seem to affect the risk of coronary artery disease in obese diabetic patients [16].

The pathophysiology of DM is associated with upregulated oxidative stress due to the elevated concentration of glucose [15]. PON1 activity is decreased in both type 1 and 2 DM [14]. Moreover, the activity level of PON1 in T2DM inversely correlates with the duration of the disease and with metabolic parameters such as plasma glucose concentration, glycated hemoglobin (HbA1c) and homeostatic model assessment of insulin resistance, which is a method for assessing β-cell function and insulin resistance based on fasting glucose and insulin or C-peptide concentrations [16,17]. Hence, the decreased activity of PON1 might contribute to DM progression. To support this hypothesis, in a subanalysis of the “Metformin, arterial function, intima-media thickness and nitroxidation in the metabolic syndrome (MEFISTO)” study, patients treated with metformin showed an enhancement of PON1 activity compared to controls [18]. However, other studies showed contradictory results regarding the concentration and activity level of PON1 in serum of diabetic patients. The contradictory results might be explained by PON1 genetic polymorphism, which may determine the variable role of PON1 in diabetes. More studies are needed to define the importance of PON1 in patients with DM in clinical practice [15,16].

DM is strongly linked to atherosclerosis, with ischemic stroke as one of the most common complications. A negative correlation between PON1 activity and the risk of stroke was observed, likely due to the antiatherogenic properties of PON1. The antiatherogenic features of PON1 are related to its role in inhibition of lipid oxidation, breakdown of lipid peroxides by hydrolysis and prevention of accumulation of lipid peroxides in low-density lipoproteins. The Q192R allele of PON1 seems to be the most promising predictor of stroke [15,19]. Determining the association between the alleles of PON1 and DM-related complications including stroke is an interesting scope of future research.

Neurodegenerative diseases such as AD and PD are also associated with oxidative stress and dysregulated acetylcholine metabolism [19]. Decreased PON1 activity impairs organophosphate detoxification and leads to acetylcholine accumulation and disrupted neurotransmission. However, the relationship between PON1 activity and the above-mentioned neurodegenerative diseases is not yet clear. A meta-analysis of 15 studies showed a promising relationship between highly specific genotypes of PON1 polymorphism with AD in the Caucasian population [15]. However, other studies showed no association between PON1 polymorphism and AD risk. Similarly, although a potential link between a few alleles of PON1 and PD remains, the dominating outcomes present no significant relation between PD and PON1 polymorphisms [19]. The ambiguity of outcomes might emerge from the differences in methodology and study group ethnicity. More research is needed to provide firm conclusions of the role of the PON1 as a biomarker in neurodegenerative diseases in DM.

### 2.3. Tau, β-Amyloid and Glycogen Synthase Kinase 3β

Tau is a microtubule-associated protein present mainly in the brain, responsible for basic cellular functions, such as cell morphogenesis, cell division and intracellular transport [20]. In addition, tau protein is primarily connected to the pathogenesis of AD. Likewise, amyloid beta peptide (Aβ) is commonly known to form the amyloid plaque on nerve cells found in brains of patients with AD [95]. Research suggests that tau phosphorylation and Aβ accumulation both play a role in AD development [21].

There is also an association between tau and diabetes, since both insulin and IGF-1 are involved in tau phosphorylation, which is associated with neurofibrillary tangles production and synaptic loss. Tau phosphorylation starts when the insulin signaling in the brain is impaired, resulting in decreased Akt kinase activity (also known as protein kinase B) and increased activity of glycogen synthase kinase 3 beta (GSK-3β) [22]. GSK-3β is an essential regulator of glucose blood level by participating in glycogen synthesis [20]. Moreover, GSK-3β plays a role in the pathogenesis of insulin deficiency and insulin resistance, both of which are crucial in DM development [23]. A recent study showed that GSK-3β activity is upregulated in cognitively impaired T2DM patients, in comparison with DM patients with no cognitive impairment, thus confirming the association between GSK-3β peripheral activity and cognitive impairment in T2DM [96]. Further, increased levels of total tau and phosphorylated tau in CSF were found both in type 1 and 2 DM patients, compared to controls [97,98]. In addition, the concentration of Aβ peptide 42 (Aβ42) in CSF was higher in type 1 DM patients, compared to healthy controls [88]. Finally, dysregulated glucose metabolism in DM leads to formation of advanced glycation end products (AGE). Higher levels of AGE in the brain have been found to promote Aβ42 aggregation by hindering its removal. AGE receptor density is increased in AD and involved in the Aβ-related inflammatory processes [24]. To support the link between T2DM and neurodegeneration via tau and Aβ, results of several clinical studies confirmed that long-term use of metformin in diabetic patients contributes to better cognitive function compared to using other anti-diabetic drugs. Metformin was found to decrease tau phosphorylation and Aβ production by reducing the activity of acetylcholinesterase and subsequently increasing the brain concentration of acetylcholine, a neurotransmitter involved in the process of learning and memory [99].

Altogether, it is likely that the commonly used AD biomarkers (increased GSK-3β activity, tau protein hyperphosphorylation and Aβ accumulation) are associated with memory impairment in patients with diabetes, suggesting the potential application of these biomarkers in DM patients. Yet, more studies are needed to draw firm conclusions.

### 2.4. Phosphoinositide 3-Kinases

PI3K are signaling proteins which participate in maintaining glucose homeostasis. PI3K defect increases serum glucose concentrations, implicating their role in the pathophysiology of DM [25]. Moreover, PI3K mediates antiatherogenic and vasodilatory effects of insulin by endothelial nitric oxide synthase (eNOS) activation and nitric oxide (NO) production, which are responsible for the insulin-related vascular effects [25]. Due to PI3K antiatherogenic and vasoprotective mechanisms, their increased expression might potentially lower the risk of vascular complications in DM patients.

One of the neurological complications in DM is hypothalamus–pituitary–adrenal axis hyperactivation, which leads to cognitive impairment. Increased PI3K activity reversed this impairment in mouse and rat models [25]. It was also observed that peroxisome proliferator-activated receptor γ agonist (rosiglitazone) improved cognitive powers via a PI3K-dependent activation mechanism [25]. Therefore, decreased PI3K activity could be a potential predictor of neurodegenerative disorders in DM, and a potential therapeutic target to reverse it.

PI3K also plays a role in AD development, since it is the first element of the insulin/PI3K/Akt/mTOR (mammalian target of rapamycin) pathway. PI3K enhances the phosphorylation of tau protein and interferes with Aβ, which inhibits this pathway in neuronal (stem) cells, causing neurotoxicity [26]. Finally, decreased PI3K activity has been observed in the central nervous system in T2DM patients with AD, potentially linking DM with neurodegenerative disorders [96]. Concurrently, downregulation of PI3K activity in the peripheral nerves is linked to decreased retrograde transport of neurotrophins and nerve growth factor (NGF). In addition, PI3K plays a key role in nerve survival. Altogether, reduced PI3K activity seems to participate in the development of diabetic peripheral neuropathy (DPN) [25].

Finally, the insulin/PI3K/Akt/mTOR pathway promotes the survival of dopamine-producing neurons by inhibiting apoptosis. This pathway is downregulated in the brains of PD patients, thus impeding the survival of dopaminergic neurons, which suggests that the PI3K-initiated pathway is involved in the development of PD as well [26]. To support this notion, metformin increases the levels of mTOR in substantia nigra where dopaminergic neurons are located, explicitly linking DM and PD [100].

Altogether, it seems that PI3K has a protective role in DM and in neurodegenerative disease, and that there might be an association between the decreased levels of PI3K in both pathophysiological states. The association between DM and neurodegeneration mediated via the insulin–PI3K–Akt signaling pathway is shown in Figure 2.

### 2.5. Amylin

Amylin is a pancreatic β-cell peptide hormone, which is released together with insulin and regulates food intake by controlling glucose homeostasis [27]. Amylin has a mainly anorectic role by decreasing the secretion of gastric acid and glucagon [28]. Amylin secretion is upregulated following increased consumption of carbohydrates and fats, which leads to amylin aggregates formation [27]. These aggregates induce the injury of cellular organelles including mitochondria, leading to reactive oxygen species generation and inflammation with subsequent damage and loss of various cell types, including pancreatic β-cells [28]. The cytotoxic effect of amylin aggregates on pancreatic β-cells strongly links amylin to T2DM. Hyperamylinemia is often present in patients with pre-diabetic insulin resistance [28,29], and amylin deposits were found in the pancreas of 95% of T2DM patients, classifying DM as an amyloid disease [27,29]. Worthy of note, amylin may cross the brain blood barrier and promote mixed plaques formation, consisting of amylin aggregates and Aβ [30]. These plaques have recently been identified in brain tissue of T2DM patients suffering from cognitive impairment [28]. Similarly, amylin aggregates were found in the grey matter of T2DM patients’ temporal lobes, known to participate in perception and memory [28]. Concurrently, the amylin analogue pramlintide was shown to improve cognitive impairment in AD, suggesting that administration of exogenous amylin-type peptides have the potential to become a new therapeutic avenue for AD [31].

Altogether, amylin could be considered as a potential link between T2DM and neurodegenerative disease, such as AD, and used as a potential biomarker for neuronal damage in DM patients [30]. Further studies are needed to determine the association between the concentrations of amylin in peripheral blood and neurodegeneration and a randomized, double-blind, placebo-controlled clinical trial is required to examine the efficacy of the amylin analogue pramlintide for AD.

### 2.6. Dopamine

Dopamine is a neurotransmitter which plays a key role in multiple brain functions including learning, motor control and executive functions. Alterations in dopaminergic signaling participate in the development of neurodegenerative diseases, especially PD [32]. Dopamine also has other functions, including inhibition of prolactin secretion, appetite regulation and downregulation of insulin secretion in pancreatic β-cells [33]. Concurrently, insulin crosses the blood brain barrier and affects dopamine activity by insulin receptors in the dopaminergic neurons of the midbrain [101]. Insulin upregulates dopaminergic transport and boosts the elimination of dopamine from the synapse, therefore affecting dopamine neurotransmission [101]. Hyperglycemia, in turn, decreases the insulin level in the central nervous system [34], which further supports the link between DM and PD. In addition, insulin resistance—a characteristic component of T2DM—has been found in brains of patients suffering from PD. Insulin resistance also correlates with the progression of PD [35]. Finally, a recent study concluded that DM can possibly increase the risk of development of Parkinson-like pathology and aggravate the disease phenotype [102].

Dopamine downregulation is a major factor in the development of diabetic complications, including DR [36,101,103]. Dopamine is the essential neurotransmitter in the retina [36]. Chronically disturbed glucose homeostasis in the retina results in hyperglycemia and leads to neuronal damage through the decline of dopamine production, linking hyperglycemia with DR [36]. In PD patients, loss of dopaminergic neurons is followed by inner retinal thinning [37]. Altogether, dopamine represents a potential biomarker for neurodegenerative complications is DM, but potentially also for DM development in neurodegenerative disease.

### 2.7. Gamma-Glutamyl Transferase

GGT is a key player in glutathione metabolism, which is an essential cellular antioxidant [38]. GGT is a commonly used diagnostic test to assess liver function [104]. Yet, its role may be underestimated in pathologies of other systems, including diabetes and neurodegenerative diseases [39]. In diabetic patients, higher plasma levels of GGT have been observed compared to healthy controls [40]. Moreover, the increased levels of GGT correlate with the risk of pre-diabetes and T2DM [41]. In a meta-analysis of six studies including 4726 patients with T2DM, thiazolidinediones significantly reduced the alanine transaminase, aspartate aminotransferase and GGT plasma levels, compared with metformin, with pioglitazone exerting the most potent hepatoprotective effect [29]. The mechanism underlying the relationship between T2DM and GGT may be associated with upregulated oxidative stress and lipid accumulation in the hepatocytes, which leads to insulin resistance in the hepatocytes and affects the development of T2DM [40]. High concentrations of GGT are also observed in DR, which makes GGT a possible biomarker to predict and/or diagnose diabetic complications [42].

Oxidative stress has fundamental effects on neuronal damage and thus on neurodegenerative diseases, such as dementia [43]. In a recent study, higher GGT variability and baseline GGT concentration in DM patients were two independent factors enhancing the risk for dementia, both AD and vascular dementia [44]. GGT and its fluctuations are directly engaged in plaque progression and associated with the decline of cognitive function [44]. Altogether, GGT may play a role in predicting dementia development in patients with DM.

PD is another neurodegenerative disorder where GGT might be involved. Higher levels of GGT activity were correlated with lower risk of PD development in men but, on the contrary, were higher in women [94]. The suggested underlying mechanism seems to originate in oxidative stress and neuroinflammation, resulting from abnormal protein aggregation and mitochondrial dysfunction in the substantia nigra in the course of PD [45]. In addition, GGT participates in PD development through impairment of glutathione metabolism with subsequent neuronal cell death.

One of the most common complications of DM is DPN, which is also associated with GGT serum level disturbances [46]. Metabolic and vascular pathways accompanying oxidative stress may initiate and trigger the progression of neuronal damage in diabetic neuropathy [95]. Indeed, DM patients with polyneuropathy presented with higher levels of serum GGT compared to patients without polyneuropathy [105], implying the existence of a GGT-associated network between diabetes and its complications.

Last but not least in importance, GGT levels seem to affect the risk of stroke. The positive relationship between GGT concentrations and stroke were presented in a few independent studies [47,104]. Furthermore, a meta-analysis including 5707 stroke cases among 926,497 participants in 10 prospective studies stated that GGT level is an independent stroke risk factor [47]. Moreover, GGT is present in atherosclerotic plaques and linked with higher prevalence of cardiovascular incidents and coronary calcifications [48]. Therefore, GGT found in calcified intracranial atherosclerotic plaques might indicate greater risk for stroke [49]. Altogether, GGT emerges as a possible marker to predict stroke. However, its association with stroke in patients with DM needs to be elucidated.

### 2.8. Growth Factors

GFs regulate the physiological growth, maturation and repair of all tissues, and are also affected by pathological states [50]. For example, long-term hyperglycemia causes the abnormal expression of some GFs, including vascular endothelial growth factor (VEGF), transforming growth factor β (TGF β) and platelet-derived growth factors (PDGF) [50]. Abnormal expression of GFs is found in various diabetic neurovascular and neurodegenerative complications, and metformin inhibits the development of diabetic retinopathy by inducing alternative splicing of VEGF [49].

One of the growth factors that has recently gained popularity is epidermal growth factor (EGF). It is produced by various cell types (e.g., platelets, macrophages, fibroblasts) and participates in proliferative and wound healing processes [51]. Some studies showed that EGF may be linked to the development of microvascular complications in diabetes [51]. For example, patients with type 1 DM and microangiopathy have higher serum EGF concentrations compared to those without [102]. VEGF, in turn, is mainly responsible for stimulating endothelial cell proliferation and migration, collagen production and macrophage chemotaxis [52]. Chronic hyperglycemia upregulates the expression of the VEGF family member, VEGF-A, which was found elevated in serum and urine of DM patients [106].

The relationship between GFs and DR seems to be the most promising. It has been demonstrated that concentrations of different GFs are associated with retinal ischemia, which results in neovascularization [53]. VEGF, PDGF and basic fibroblast growth factor (bFGF) are elevated in vitreous fluid of patients with DR, which supports this relationship [50]. VEGF has been associated with increased permeability of the blood–retina barrier and upregulation of neovascularization, which are often found in DR [52]. Another study proposed hypoxia and hyperglycemia as the main pathologies underlying the development of DR [54,55]. First, VEGF participates in the development of vascular pathologies, and higher levels of VEGF were found in type 1 DM children and adolescents with DR [52]. Hence, it was suggested that VEGF may be a biomarker of early stages of DR, which cannot yet be diagnosed with currently available methods [52]. Second, increased levels of VEGF were found in the vitreous body fluid of patients with proliferative diabetic retinopathy (PDR), and differences in VEGF concentration were found between patients with PDR and non-proliferative diabetic retinopathy (NPDR) [107]. Altogether, the level of VEGF in the vitreous body fluid may reflect the progression of DR, suggesting the potential of VEGF as a biomarker [56,57].

Another growth factor related to DR is pigment epithelium-derived factor (PEDF). PEDF is produced by a variety of ocular cell types and participates in the inhibition of angiogenesis. Microvascular dysfunction of endothelial cells in retina, which results from oxidative stress and inflammation in the course of diabetes, can be restored with exogenous PEDF [58]. Serum levels of PEDF are elevated in type 1 and 2 DM, but with no specific link to DR [57]. On the contrary, PEDF concentration was decreased in the vitreous body fluid of diabetic patients with DR, compared to diabetic patients without retinopathy [52]. Hence, PEDF levels in the vitreous body seem more promising in terms of biomarker development compared to their serum levels.

TGF-β regulates cell growth and differentiation, but also angiogenesis [59]. It promotes the thickening of basal lamina of retinal capillaries, which is known to participate in the pathogenesis of DR [60]. Among diabetic patients, those with DR present with higher levels of TGF-β1 in serum in comparison with patients without retinopathy [52]. In addition, serum concentrations of TGF-β1 are over 10-fold higher in patients with T2DM and NPDR compared to healthy controls [51]. Therefore, TGF-β may potentially be used as a biomarker of DR.

Another complication of DM that may be associated with GFs is hyperglycemia-related neuropathy. The mechanism underlying diabetic neuropathy is based on dysregulated GF response to ischemia and oxidative stress [61]. NGF plays a key role in the neuronal development and function of both central and peripheral nervous system [62]. Lower concentrations of NGF are connected to reduced values of nerve conduction velocity, which is a fundamental player of the pathogenesis of DP [63].

A few studies have supported the hypothesis of decreased serum levels of NGF in diabetic patients [108]. Moreover, decreased levels of serum NGF were observed in diabetic patients and in patients with peripheral neuropathy and compared to healthy individuals [108]. In addition, in patients with DPN, decreased serum levels of NGF accompanied the progression of the disease [109]. In addition, the changes in serum levels of VEGF according to the stage of polyneuropathy were observed. Markedly lower VEGF levels have been noted in the asymptomatic phase of DPN compared to the symptomatic phase [110]. Therefore, NGF and VEGF are promising targets as potential biomarkers of diabetic neuropathy

Another important NGF factor is BDNF. BDNF is a member of the neurotrophin family, which plays a crucial role in the development of the nervous system while supporting the survival of existing neurons and instigating neurogenesis [111]. Altered levels of BDNF, both in the circulating blood and in the CNS tissues, are involved in the pathogenesis of neurodegenerative diseases, including AD, PD and dementia, as well as in ischemic stroke [64,65,66]. BDNF acts through several different pathways including the MAPK pathway and PI3K-Akt cascade, which induce cell survival and synaptic plasticity. Apart from the nervous system, many studies showed the importance of BDNF during systemic or peripheral inflammatory conditions, such as acute coronary syndrome and T2DM [67,68,111]. It is known to play an important role in glucose and energy homeostasis. Moreover, several in vitro, in vivo and human studies suggested that BDNF might be used as a potential treatment for T2DM due to its antidiabetic and antilipidemic effects. On the other hand, both peripheral and intrathecal administration of BDNF caused several adverse effects, such as sensory thresholds reduction, pain induction, promotion of tumor progression and metastasis [69,111,112].

Regarding neurodegenerative disorders, several studies suggested links between PEDF, EGF, TGF-β and PD/AD [113,114,115,116]. For example, EGF is lower in plasma of PD patients compared to healthy controls [113]. In addition, the association between TGF-α and TGF-β levels in CSF and PD has been suggested [114]. In contrast, elevated level of TGF-β was found in CSF in AD [115]. Further, in AD the serum level of PEDF was decreased compared to two groups of healthy controls: middle-aged adults and older adults [116]. Altogether, the role of GFs as biomarkers requires further investigation.

### 2.9. Homocysteine

Homocysteine is an amino-acid derived from methionine and can be converted back into methionine or cytosine with the help of B-vitamins [70]. Homocysteine has a toxic effect on neuronal and vascular endothelial cells. What is more, homocysteine is perceived as a key player in brain damage and cognitive and memory decline [71]. The underlying mechanism is based on the state of oxidative stress, which results from homocysteine suppressing production of nitric oxide by endothelial cells and provoking formation of reactive oxygen species. Furthermore, homocysteine impedes activity of glutathione peroxidase, which leads to enhanced proliferation of endothelial cells [72]. Moreover, the metabolism of homocysteine is strongly connected to insulin levels [73]. It has been confirmed that dysregulated insulin levels in diabetes may impair the metabolism of homocysteine, resulting in elevated levels of this amino-acid [73]. Accordingly, T2DM patients and pre-diabetic patients have higher levels of plasma homocysteine [72,74]. However, in a recent meta-analysis of 12 studies involving 1156 participants, the overall effect of metformin on the concentration of serum homocysteine was neutral, suggesting that the anti-inflammatory effects of metformin demonstrated in numerous other studies are not exerted via decreased homocysteine concentration [74].

Hyperhomocysteinemia leads to endothelial dysfunction and stimulates platelet activation, which may increase the risk of atherothrombotic incidents, including ischemic stroke [72,75]. Increased concentration of plasma homocysteine was associated with carotid intima–media thickness (IMT). Consequently, IMT was higher in diabetic patients with stroke compared to patients with stroke but no DM [76]. Therefore, homocysteine plasma levels are implied for use as a potential biomarker for prediction of ischemic stroke [75]. Many studies confirmed the relationship between risk of stroke and hyperhomocysteinemia [77,117]. Moreover, several studies suggested that the differences in homocysteine plasma levels could also predict the subtype of stroke. However, further studies are needed to apply these results in clinical practice [118].

Noteworthily, a correlation between homocysteine plasma level and intracranial aneurysm formation has been found [119]. Patients with intracranial aneurysms had substantially higher mean serum homocysteine levels compared to the control group [120]. Whether the elevated serum level of homocysteine indicates the risk of intracranial aneurysm development remains to be established [119].

Recently, the relationship between hyperhomocysteinemia and diabetes-induced microangiopathies has gained a lot of attention [120]. The potential mechanisms underlying this relationship include homocysteine-induced oxidative stress or increased levels of VEGF [72]. Homocysteine has also been found to affect the blood–retina barrier and also provoke the death of retinal ganglion root cells [71,72]. Consequently, the concentration of homocysteine is increased both in plasma and in the vitreous body in DR compared to healthy controls [76,121]. In addition, differences between plasma levels of homocysteine in diabetics with PDR, NPDR and diabetics without retinopathy were found, with the highest level in PDR compared to other groups [57]. Hyperhomocysteinemia is also associated with micro- and macrovascular changes, which could lead to nerve injury, and elevated levels of plasma homocysteine are an independent risk factor of diabetic neuropathy [78]. Hence, homocysteine could potentially be used as a marker for DR diagnosis and progression monitoring, or a marker of diabetic neuropathy [78].

Homocysteine is a promising biomarker indicating not only neurovascular impairments but also neurodegenerative diseases like AD, PD and dementia. The impact of homocysteine on the neurodegenerative disorders may rise from the deficiency of cofactors related to homocysteine metabolism (B-group vitamins), drugs, diseases (e.g., renal failure) or age [71]. Homocysteine neurotoxicity may also directly influence the process of neurodegeneration, or stimulate Aβ accumulation in the brain, and promote calcium influx, apoptosis and death of neuronal cells, which lead to dementia [71]. It was shown that patients with elevated plasma homocysteine levels were more prone to develop dementia, MCI and AD in comparison to patients with normal levels of plasma homocysteine [71,79]. In AD, the initial plasma homocysteine level correlated with the severity of cognitive impairment [71]. Further, several studies confirmed hyperhomocysteinemia in PD patients with higher levels of plasma homocysteine in PD than in AD and MCI [74,80]. The underlying mechanism could be related to the neurotoxic effect of homocysteine on dopaminergic neurons in the substantia nigra [80].

Altogether, plasma homocysteine concentration is a promising biomarker for neurovascular and neurodegenerative complications of diabetes. However, hyperhomocysteinemia was also noted in elderly patients who were otherwise healthy. In addition, homocysteine levels were not yet found to be connected with cognitive function of T2DM patients [122]. Therefore, future studies on homocysteine as a biomarker should carefully take into account the potentially confounding factors.

## 3. Conclusions

In conclusion, despite extensive research on biomarkers of DM and its complications, the association between DM and neurovascular and neurodegenerative disorders requires further studies. The most promising biomarkers of neurovascular and neurodegenerative complications in DM include hsCRP, miRNAs, GGT and homocysteine.

Many studies connect the pathogenesis of diabetes-related complications with inflammatory response, oxidative stress and vascular dysfunction. Taking into account the broad availability and cost-efficacy of hsCRP, it could be a good screening biomarker for neurovascular and neurodegenerative complications in DM. The wide range of miRNAs opens the possibilities for their use in many diabetic complications but also limits their target specificity. With the plethora of ongoing studies focusing on miRNAs, new biomarkers based on miRNAs are likely to be found. GGT might potentially be used to predict such diabetic complications like DPN, DR, cognitive function impairment or stroke. However, fluctuations of GGT serum level seem to be more promising for assessing the diabetic complications than a single GGT measurement, implying that serial measurements might be needed. Serum homocysteine levels are increased in the majority of neurovascular and neurodegenerative complications of DM, including AD, MCI, dementia and DR and correlate with their progression, but other patient’s characteristics such as age and comorbidities should be taken into account when developing a homocysteine-based biomarker.

Other biomarkers described in this review (amylin, Aβ, dopamine, GSK-3β, PON-1, PI3K, tau protein and GFs) have substantial limitations, because they are either associated with only a few diabetic complications, or the conclusions regarding their role are insufficient. However, these biomarkers could become a part of a multi-biomarker screening panel for diabetic patients at risk of neurovascular and neurodegenerative complications. Such a panel might be a novel tool for individual risk stratification and early prevention of these complications in DM, allowing one to tailor the treatment strategy according to the specific patient’s needs.

## Figures and Tables

**Figure 1 jcm-09-02807-f001:**
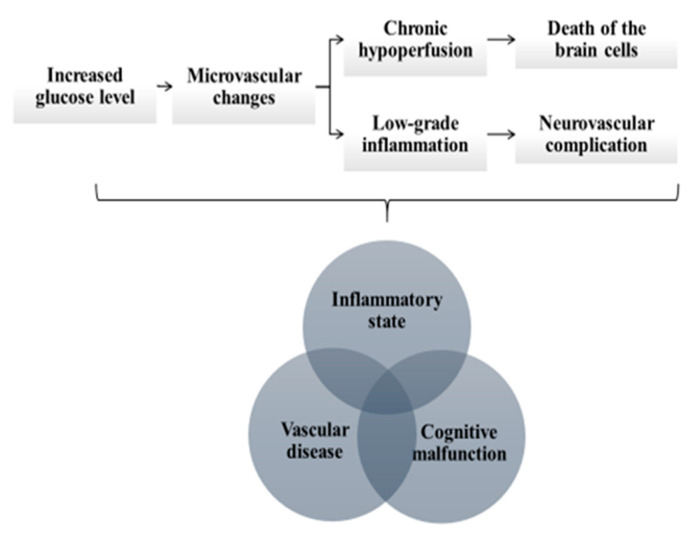
The pathophysiological mechanisms linking diabetes with neurodegenerative and neurovascular disorders include (i) chronic inflammatory state, (ii) vascular disease and cognitive malfunction. The permanently increased glucose level leads to microvascular changes, which causes chronic hypoperfusion of the brain and subsequent degeneration and death of the brain cells. The microvascular changes also trigger low-grade inflammatory response and further aggravate the risk of neurodegeneration and neurovascular complications.

**Figure 2 jcm-09-02807-f002:**
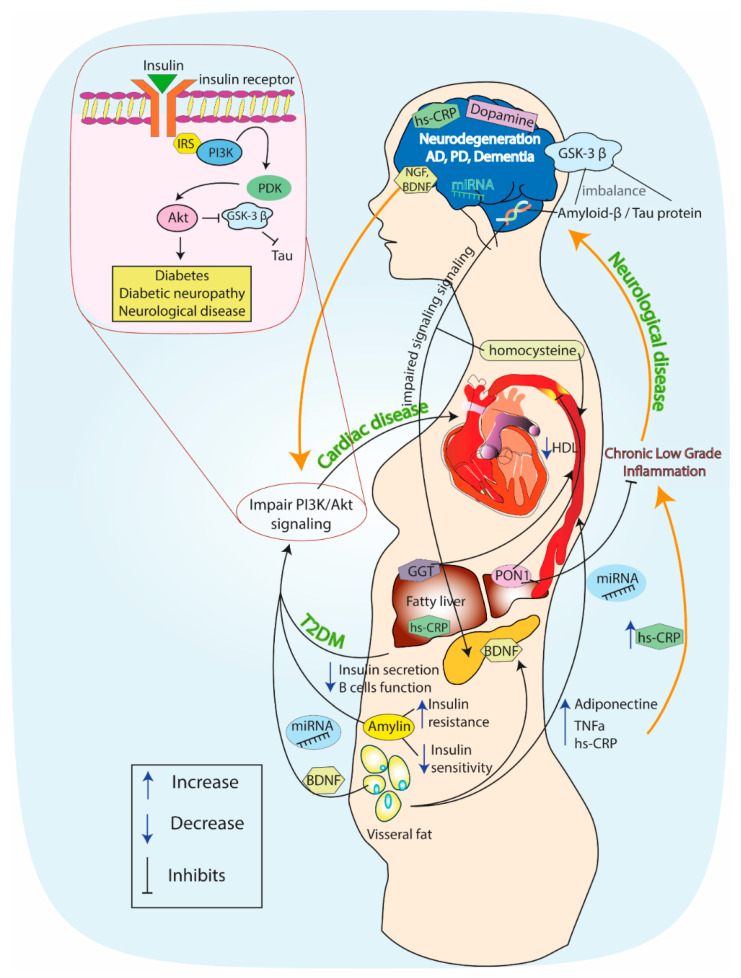
The association between diabetes mellitus and neurodegeneration mediated via the insulin–phosphoinositide 3-kinase (PI3K)–Akt signaling pathway. Description in the text.

**Table 1 jcm-09-02807-t001:** Diabetes-related neurodegenerative and neurovascular disorders, which are the focus of this review.

Neurodegenerative Diseases	Neurovascular Disorders
Alzheimer’s disease	Diabetic retinopathy
Parkinson’s disease	Diabetic neuropathy
Mild cognitive impairment	Stroke
Dementia	

**Table 2 jcm-09-02807-t002:** Associations between biomarkers discussed in this review and diabetes-related neurodegenerative and neurovascular disorders.

Biomarker	Role/Effect in Human Body	Link with T2DM	Link with Neurological Complications	Example of Neurological Complications	Direction of Change	Ref
CRP	acute phase protein, produced in response to inflammation and interleukin IL-6	increased glucose concentration →inflammation →microvascular changes	microvascular changes →chronic brain hypoperfusion → degeneration of the brain cells	AD, MCI, PD	↑	[6,7,8,9,10,11]
			microvascular changes →choroidal endothelial cell dysfunction	AMD	↑	
PON-1	antiatherogenic, antioxidant and anti-inflammatory properties (inhibition of lipid oxidation, breakdown of lipid peroxides)	increased glucose concentration → enzymatic glycation and oxidative stress	oxidative stress → endothelial damage → atherosclerosis in the brain arteries	stroke	↓	[14,15,16,17,18,19]
			oxidative stress → dysregulated acetylcholine metabolism and organo-phosphates detoxification	AD, PD	↓	
GSK-3β, tau protein, amyloidβ	morphogenesis, cell division and intracellular transport	impaired insulin signaling → increased GSK-3β activity →Increased AGE formation	increased AGE formation → tau hyperphosphorylation, Aβ accumulation → amyloid plaque aggregation on nerve cells → synaptic loss	AD, PD	↑	[20,21,22,23,24]
PI3K	glucose homeostasis, antiatherogenic and vasodilatory effect, e-NOS activation and NO production	impairment of insulin/PI3K/Akt signaling → alteration of mTOR signaling	alteration of mTOR signaling → tau hyper-phosphorylation, Aβ accumulation → neurotoxicity	AD, PD	↓	[25,26]
	key role in nerve survival	impairment of insulin signaling → decreased PI3K activity in peripheral nerves	decreased PI3K activity → decreased retrograde transport of neurotrophins and nerve growth factor	diabetic neuropathy	↓	
	promotion of the survival of dopamine-producing neurons	impairment of insulin signaling →decreased PI3K activity in CNS	decreased PI3K activity → dopamine neuron degeneration	PD	↓	
Amylin	glucose homeostasis, decrease of secretion of gastric acid and glucagon	amylin aggregates → cytotoxic effect on pancreatic β-cells	amylin aggregates →mixed plaque formation (amylin, β-amyloid)	AD, MCI	↑	[27,28,29,30,31,32]
DA	glucose homeostasis motor control executive functions	upregulated insulin activity through insulin receptors → modulation of dopaminergic neurons in midbrain	modulation of dopaminergic neurons →boost of DA elimination from the synapse; degeneration of dopaminergic neurons	PD	↓	[33,34,35,36,37]
	neuro-transmitter in retina	chronically disturbed glucose homeostasis	disturbed glucose homeostasis → decline of DA production → neuronal damage	DR	↓	
GGT	glutathione metabolism	oxidative stress →insulin resistance	upregulated oxidative stress and lipid accumulation in the retina → retinopathy	DR	↑	[38,39,40,41,42,43,44,45,46,47,48]
	cellular antioxidant	increased glucose concentration → oxidative stress	oxidative stress → neuronal damage decline of cognitive function, plaque progression	AD, dementia	↑/↓^1^	
	glutathione metabolism, cellular antioxidant	increased glucose concentration → oxidative stress, neuro-inflammation	toxic abnormal protein aggregation and mitochondrial dysfunction in substantia nigra → oxidative stress, neuro-inflammation → neuronal cell death	PD	↑	
	cellular antioxidant	increased glucose concentration → oxidative stress	oxidative stress → progression of neuronal damage in diabetic neuropathy	diabetic neuropathy	↑/↓^1^	
	cellular antioxidant, independent risk factor of stroke	increased glucose concentration → oxidative stress	oxidative stress → GGT presence in calcified intracranial atherosclerotic plaques	stroke	↑	
GF						[49,50,51]
EGF	proliferative and wound healing processes	increased glucose concentration → inflammation →microvascular complications	microvascular complications → retinal ischemia	DR	↑	[50]
VEGF	endothelial cell proliferation and migration, collagen production, macrophage chemotaxis	increased glucose concentration → inflammation →microvascular complications			↑	[49,51,52,53,54,55]
PEDF	angiogenesis inhibition	increased glucose concentration →oxidative and inflammatory conditions	oxidative and inflammatory conditions → retinal microvascular endothelial cell dysfunction	DR	↓	[51,56,57]
TGF-β	cell growth and differentiation, angiogenesis	increased glucose concentration →oxidative and inflammatory conditions	oxidative and inflammatory conditions →thickening of basal lamina of retinal vessels	DR	↑	[51,58,59]
NGF	neuronal development, growth and survival of neurons in the nervous system	glucotoxicity, insulin deficiency →ischemia and oxidative stress	ischemia and oxidative stress → decreased nerve conduction velocity	peripheral neuropathy	↓	[60,61,62,63]
BDNF	supporting the survival of existing neurons, neurogenesis instigation	dysregulation of glucose level → impairment of insulin signaling →altered MAPK and PI3K activity	altered MAPK and PI3K activity → potential role in stroke recovery (plasticity promotion)	ischemic stroke	↓	[64,65,66,67,68,69]
Homo-cysteine	insulin homeostasis, suppression of endothelial NO production, increase of ROS	increased glucose concentration → oxidative stress	oxidative stress → endothelial dysfunction→ platelet activation, atherothrombotic incidents, carotid intima–media thickness	stroke	↑	[70,71,72,73,74,75,76,77,78,79,80]
	activation of inflammatory and oxidative stress mechanisms	increased glucose concentration → oxidative stress; increased levels of VEGF → micro-angiopathy	micro-angiopathy → direct effect on blood–retina barrier, apoptosis in retinal ganglion cells	DR	↑	
	independent risk factor	increased glucose concentration→inflammation →micro and macrovascular complications	inflammation → micro and macrovascular complications → nerve injury	diabetic neuropathy	↑	
	role in brain damage, cognitive and memory decline, activation of oxidative stress mechanisms	increased glucose concentration → oxidative stress → ROS production	deficiency of cofactors related to homocysteine metabolism → HCY neurotoxicity → ROS production → Aβ accumulation in the brain →apoptosis and neuronal death	AD, PD, dementia	↑	

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
