# Peer review of "Early Biomarkers of Neurodegenerative and Neurovascular Disorders in Diabetes"

_jcm, 2020, doi:10.3390/jcm9092807_

Round 1

Reviewer 1 Report

In their review, Aleksandra Gasecka et al. present an interesting overview of the early biomarkers being dysregulated in diabetes and neurodegenerative / neurovascular disorders. The introduction is clear, providing the biomarkers listing, then each of them is described in different paragraphs by showing their connections to any diabetic type or neurodegeneratives disorders. Interestingly, the authors introduced well-known diabetic (C-Reactive Protein, gamma-glutamyl transferase, glycogen synthase kinase 3ß, phosphoinositide 3-kinases, growth factors) or neurodegenerative (dopamine, tau protein, ß amyloid) biomarkers, but also new ones (homocysteine, microRNAs, paraoxonase 1,amylin), highlighting a well-documented work.

The review provides a table (table 1) and two clear figures (fig 1 and Fig 2) summarizing and integrating the different effects of each biomarker.

Minor Comments

Line 36, What is the meaning of (vii)?

Line 37, the authors introduced arylesterase 1 without mentioning it anymore in the review.

Figure 2: it would be interesting if the authors could integrate all the markers they mentioned in the review: ß amyloid, tau protein, glycogen synthase kinase 3ß, homocysteine, micro-RNAs, paraoxonase 1 and the growths factors are missing.

Author Response

Dear Reviewer,

we are thankful for the time and effort that you spent to provide in-depth review of our manuscript. We corrected our manuscript according to your requests. Our response and corrections are listed below.

Minor Comments

Line 36, What is the meaning of (vii)?

Thank you for noticing this, (vii) is a typo from the previous version of the abstract. We removed it.

Line 37, the authors introduced arylesterase 1 without mentioning it anymore in the review.

Thank you for this remark, we removed the term "arylesterase" and consequently used the term "paraoxonase" throughout the manuscript.

Figure 2: it would be interesting if the authors could integrate all the markers they mentioned in the review: ß amyloid, tau protein, glycogen synthase kinase 3ß, homocysteine, micro-RNAs, paraoxonase 1 and the growths factors are missing.

We are grateful for this suggestion, we modified figure 2 and included there all markers mentioned in the review.

In summary, we appreciate your time and comments. We hope that the present version of the manuscript will be acceptable for publication.

Best regards,

Aleksandra Gasecka

Reviewer 2 Report

The review focused on biomarkers of diabetes-related neurodegenerative and neurovascular disorders. There have been several studies reported of association between DM and neurovascular and neurodegenerative disorders like diabetic retinopathy, diabetic neuropathy, stroke, Alzheimer’s disease, Parkinson’s disease and dementia. Therefore, identification of new disease specific biomarker has always been the main focus of research specially involving at early stage of disease. This review is a valuable addition towards identification of such biomarkers.

Some minor spelling and grammar checks needs to be done.

Author Response

Dear Reviewer,

we are thankful for the time and effort that you spent to review of our manuscript. Thank you for your appreciation of our manuscript. We double checked the text to correct the spelling and grammar mistakes. We hope that the present version of the manuscript will be acceptable for publication.

Best regards,

Aleksandra Gasecka

Reviewer 3 Report

Hello,

The manuscript is well written and organized. It contains an updated information about the early biomarkers of DM and NDs. All sections are comprehensive and the figures and tables are well very clear and well presented. The associations between the biomarkers and the NDs in DM are explained and references in a proper way. However, one thing, according to this review, is missing. It would be plausible to add a section or rather to elaborate in each section about how glucose control and management of DM by medications or dietary interventions will affect those biomarkers and possibly reduce the risk for the NDs in DM.  

Author Response

Dear Reviewer,

We are thankful for the time and effort that you spent to provide in-depth review of our manuscript. We corrected our manuscript according to your request by elaborating in each section about how glucose control and management of DM by medications or dietary interventions will affect those biomarkers and possibly reduce the risk for the NDs in DM. Our response and corrections are listed below. All changes in the new version of the manuscript are highlighted in yellow.

Lines 86-93: Concurrently, various hypoglycemic agents have been found to exert anti-inflammatory effects and decrease serum concentration of hsCRP, including metformin, peroxisome proliferator-activated receptor-γ agonists, dipeptidyl peptidase-4 inhibitors, glucagon-like peptide-1 receptor agonists and insulin [10]. On the other hand, there is evidence that metformin along with other antidiabetic drugs may prevent the development of DM-associated neurodegenerative complications by balancing the survival and death signaling in cells, thus avoiding the neuronal death that defines neurodegenerative disease [11].

Lines 142-146: There is evidence that metformin affects the expression of at least 13 different miRNAs, thus supporting the development of miRNA-based therapeutic strategies against DM [28]. On the other hand, metformin has been shown to upregulate miR‐34a, which plays an important role in responding to DNA damage and protecting against neurodegeneration, potentially providing the combined anti-diabetic and anti-neurodegenerative benefits [29].

Lines 167-170: To support this hypothesis, in a subanalysis of the “Metformin, arterial function, intima‐media thickness, and nitroxidation in the metabolic syndrome (MEFISTO)” study, patients treated with metformin showed an enhance of PON1 activity, compared to controls [34].

Lines 221-227: To support the link between T2DM and neurodegeneration via tau and Aβ, results of several clinical studies confirmed that long-term use of metformin in diabetic patients contributes to better cognitive function, compared to using other anti-diabetic drugs. Metformin was found to decrease tau phosphorylation and Aβ production by reducing the activity of acetylcholinesterase and subsequently increasing the brain concentration of acetylcholine, a neurotransmitter involved in the process of learning and memory [45].

Lines 261-263: To support this notion, metformin increases the levels of mTOR in substantia nigra where dopaminergic neurons are located, explicitly linking DM and PD [48].

Lines 282-292: Concurrently, amylin analogue pramlintide was showed to improve cognitive impairment in AD, suggesting that administration of exogenous amylin-type peptides have the potential to become a new therapeutic avenue for AD [61]. (…) Further studies are needed to determine the association between the concentrations of amylin in peripheral blood and neurodegeneration and a randomized, double-blind, placebo-controlled clinical trial is required to examine the efficacy of amylin analogue pramlintide for AD.

Lines 301-303: Insulin upregulates dopaminergic transport and boosts the elimination of dopamine from the synapse, therefore affecting dopamine neurotransmission [56].

Lines 324-327: In a meta-analysis of six studies including 4726 patients with T2DM, thiazolidinediones significantly reduced the alanine transaminase, aspartate aminotransferase and GGT plasma levels, compared with metformin, with pioglitazone exerting the most potent hepatoprotective effect [70].

Lines 368-370: Abnormal expression of GFs is found in various diabetic neurovascular and neurodegenerative complications, and metformin inhibits the development of diabetic retinopathy through inducing alternative splicing of VEGF [80].

Lines 464-467: However, in a recent meta-analysis of 12 studies involving 1156 participants, the overall effect of metformin on the concentration of serum homocysteine was neutral, suggesting that the anti-inflammatory effects of metformin demonstrated in numerous other studies are not exerted via decreased homocysteine concentration [116].

We hope that the present version of the manuscript will be acceptable for publication.

Best regards,

Aleksandra Gąsecka